# ZnO Microfiltration Membranes for Desalination by a Vacuum Flow-Through Evaporation Method

**DOI:** 10.3390/membranes9120156

**Published:** 2019-11-23

**Authors:** Shailesh Dangwal, Ruochen Liu, Lyndon D. Bastatas, Elena Echeverria, Chengqian Huang, Yu Mao, David N. Mcllroy, Sangil Han, Seok-Jhin Kim

**Affiliations:** 1School of Chemical Engineering, Oklahoma State University, 420 Engineering North, Stillwater, OK 74078, USA; shailesh.dangwal@okstate.edu (S.D.); ruochen.liu@okstate.edu (R.L.); 2Department of Physics, Oklahoma State University, 145 Physical Sciences Bldg., Stillwater, OK 74078, USA; ldbastatas@gmail.com (L.D.B.); elena.echeverria@okstate.edu (E.E.); dave.mcilroy@okstate.edu (D.N.M.); 3Department of Biosystems and Engineering, Oklahoma State University, 213 Agricultural Hall, Stillwater, OK 74078, USA; chengqian.huang@okstate.edu (C.H.); yu.mao@okstate.edu (Y.M.); 4Department of Chemical Engineering, Changwon National University, Changwon-Si, Gyeongsangnam-do 51140, Korea; sangilh@changwon.ac.kr

**Keywords:** desalination, atomic layer deposition, hydrophilicity, membrane, zinc oxide

## Abstract

ZnO was deposited on macroporous α-alumina membranes *via* atomic layer deposition (ALD) to improve water flux by increasing their hydrophilicity and reducing mass transfer resistance through membrane pore channels. The deposition of ZnO was systemically performed for 4–128 cycles of ALD at 170 °C. Analysis of membrane surface by contact angles (CA) measurements revealed that the hydrophilicity of the ZnO ALD membrane was enhanced with increasing the number of ALD cycles. It was observed that a vacuum-assisted ‘flow-through’ evaporation method had significantly higher efficacy in comparison to conventional desalination methods. By using the vacuum-assisted ‘flow-through’ technique, the water flux of the ZnO ALD membrane (~170 L m^−2^ h^−1^) was obtained, which is higher than uncoated pristine membranes (92 L m^−2^ h^−1^). It was also found that ZnO ALD membranes substantially improved water flux while keeping excellent salt rejection rate (>99.9%). Ultrasonic membrane cleaning had considerable effect on reducing the membrane fouling.

## 1. Introduction

Along with technology development, industrialization that significantly contributed for welfare of mankind has also caused alarming global water demand. Water purification has been a challenging problem in the present world. There is a crucial need to improve water treatment technologies in a more energy efficient manner and alleviate stress to water resources. Membrane technology is a potential candidate for feasible and economic water purification because it does not require additives, thermal energy, and media regeneration [1,2,3]. It offers a chemically robust desalination option and improves the economics of desalination because the theoretical energy requirements are less than thermal alternatives [4,5].

Table 1 shows the literature review for conventional desalination methods (pervaporation (PV) and membrane distillation (MD)) for different membranes. Huang et al. prepared graphene oxide (GO)/polyimide (PI) hollow fiber membranes by direct spinning of GO/PI suspension via a coaxial two-capillary spinning strategy. GO/PI hollow fiber membrane was tested for desalination showing water flux of 15.6 L m^−2^ h^−1^ and salt rejection of 99.8% for desalination at 90 °C [6]. Elma et al. prepared a silica membrane using two step sol-gel method using tetraethyl orthosilicate in ethanolic solution by using nitric acid and ammonia as catalysts. The feed concentration of the synthetic NaCl solution was varied from 0.3–15 wt% at 22 °C. Water flux and salt rejection decreased with increasing salt concentration showing an average value of 9.5 L m^−2^ h^−1^ and 99.6% for 0.3 wt% feed solution [7]. 

MD has also been a widespread approach for purification of salty water which uses membranes having pore size ranging from 0.1 to 1 µm and the membrane is not wetted by process liquids [8]. Garofolo et al. used MFI-type zeolite membranes for desalination. With water permeation tests, the membranes showed water permeance of ~42.8 L m^−2^ h^−1^ bar^−1^ and ion rejection above 99% [9]. Eykens et al. deposited hydrophobic coating (Hyflon AD80×) on a commercial hydrophilic membrane with a microporous structure. The membrane showed water flux of 16 L m^−2^ h^−1^ and ion rejection above 99.5% [8]. Leaper et al. fabricated PVDF mixed matrix membranes by incorporating graphene oxide functionalized with 3-(aminopropyl) triethoxysilane (APTS) into PVDF polymer solutions. Successful functionalization of GO with APTS was confirmed with XPS and FTIR. Addition of GO and GO-APTS enhanced the membrane flux by 52 and 86%, respectively. The best performing membrane contained 0.3 wt% GO-APTS and showed water flux of 6.2 L m^−2^ h^−1^ [10]. Bush et al. used pH adjustment as a technique to reduce silica scaling on a hydrophobic polypropylene membrane for desalination. For feed pH less than 5 or higher than 11, silica scaling became negligible to 6000 mg/L of silica concentration. Water flux of 11 L m^−2^ h^−1^ and salt rejection of 99.9% was obtained [11]. 

Although conventional desalination techniques have demonstrated excellent performance in terms of salt rejections, further investigation is needed to effectively maximize flux while keeping high salt rejections for industrial application and commercialization. In some cases, the small pore sizes of membranes might restrict flux across the membrane [12,13]. Microfiltration (MF) membranes have the technical advantages including relatively large pore sizes (50–500 nm), narrow and well-defined pore size distribution, and high porosity, which will result in a higher flux. Furthermore, pore modification is an attractive approach to improve the surface hydrophilicity and render desired functionality for effectively lowering membrane fouling.

Atomic layer deposition (ALD) process is a chemical vapor deposition process that is based upon surface limiting chemical reactions between precursors and a substrate. ALD method is suitable for depositing film with controllable thickness even at sub-nanometer scale [21]. It operates on alternating, self-limiting chemical reactions between gaseous precursors and a solid surface to deposit thin films in a layer-by-layer fashion [22]. Because the self-terminating surface reactions along with gaseous diffusion of precursor molecules induce conformal and uniform coating, ALD is well suited for catalyst synthesis and surface fabrications [23,24,25]. ALD with metal oxides like ZnO and Al_2_O_3_ has shown to increase the hydrophilicity of the membrane surface, which makes ALD useful for water desalination purpose [26,27,28,29]. Alam et al. grew TiO_2_ ALD on polyethersulphone (PES) membrane. It was shown that the TiO_2_-film-deposited PES membrane exhibited >90% NaCl rejection (4 times higher than uncoated membranes) in a pressurized desalination at feed pressure of 8 atm and room temperature. However, deposition of TiO_2_ resulted in marginal decrease in water flux from 60 to 47 L m^−2^ h^−1^ because of pore size reduction caused by nanolayer deposition [28]. Song et al. used molecular layer deposition (MLD) as a novel and highly controllable method to prepare TiO_2_ nanofiltration membranes with ~1 nm pore size for water purification. Membrane pore sizes were controlled by number of deposition cycles and precursor species. Optimized TiO_2_ nanofiltration membrane had water permeability of ~40 L m^2^ h^−1^ bar^−1^ for pressurized desalination. Membrane showed moderate rejection for Na_2_SO_4_ (43%), and MgSO_4_ (35%) and high rejection of methylene blue (96%) [30].

There is a critical need to overcome the limitation of water flux for desalination. In this work, ZnO was deposited via ALD on macroporous α-alumina membranes for purification of salty water. In order to investigate methods to increase the water flux, we conducted desalination using ZnO ALD membranes by a vacuum-assisted ‘flow-through’ evaporation method. Due to a relatively larger membrane pore size, there was bulk transport of water across the membrane. Then there was successive water evaporation in the permeate side, which purified salty water. Both bulk movement of water (across the membrane) and rapid water evaporation (on top of the permeate side) resulted in extremely high water flux. ZnO ALD growth on α-alumina membranes significantly helped in enhancing water flux and ion rejection values for desalination. It was found that the water flux across ALD membranes increased with successive deposition of ALD layers due to enhanced surface hydrophilicity. The ZnO ALD membranes showed high water flux of >150 L m^−2^ h^−1^ and high salt rejection of >99.5%.

## 2. Materials and Methods 

### 2.1. Atomic Layer Deposition Process

A thin layer of ZnO was coated on the α-alumina membranes (1 in. diameter, 1 mm thickness, ~200 nm pore size, and ~25% porosity from Coorstek) by ALD. The ZnO film was formed through the following mechanism [31]:Zn(C_2_H_5_)_2_ + H_2_O (g) → ZnO + 2C_2_H_6_ (g)(1)

The procedure was carried out using the ALD unit (OkYay Tech, Ankara, Turkey). During the ALD process, the chamber was maintained at 170 °C with a baseline pressure of ~200 millitorr. To achieve the homogeneous coating, membranes were initially stabilized inside the chamber for 30 min prior to deposition. Using N_2_ as the gas carrier, sequential dosing of DI water and diethylzinc, Zn(C_2_H_5_)_2_, (Strem Chemicals Inc., >95%, Newburyport, MA, USA) was then conducted at different number of cycles (4, 8, 16, 64, and 128) to coat a thin layer of ZnO with a film thickness of ~2 Å per cycle. In between the dosing of each precursor, the chamber was purged with N_2_ to ensure that the precursors did not react in vapor phase but rather at the surface of the alumina membranes. Figure 1 shows the schematic diagram of the ALD process.

### 2.2. Membrane Desalination Tests 

The set-up for vacuum-assisted flow-through desalination process is illustrated in Figure 2. The ZnO ALD membrane was attached to a bell-shaped glass tube with an epoxy adhesive, and with the ALD treated surface facing out of the bell-shaped tube. The membrane was immersed in the salty solution with the ALD treated surface facing the salty solution. Pressure difference between the two sides of the membrane creates the driving force for bulk motion of salty water across the membrane. The other side of the glass tube was connected to the vacuum pump (1400B-01, WELCH-ILMVAC, Denver Gardner, Laguna Hills, CA, USA) which maintained vacuum in the permeate side and the pressure differential of ~1 atm across the membrane. The pressure differential across the membrane was monitored by a pressure gauge. All the equipment (vertical glass tube, cold trap and vacuum pump) was connected with plastic tubing, and the vacuum tightness of the assembly and all connections was verified. The salty solution (0.3 wt%) in the feed side was prepared by mixing 3 g of salt (Sigma Aldrich, St. Louis, MO, USA) in 1000 g of DI water. The weight percentage of different ions in the salty water solution was Na^+^: 0.08, Mg^2+^: 0.01, S^2+^: 0.02, Cl^−^: 0.15, K^+^: 0.003, and Ca^2+^: 0.003. All desalination experiments were performed for ~1 h at room temperature. Water passed through the membrane in liquid form and then due to vacuum on the permeate side, evaporation takes place at room temperature. Evaporation at the permeate side of the membrane caused the purification of the salty water, leaving extremely salty water on top of the permeate side of the membrane. The evaporated water was then condensed in the cold trap and the amount was measured for calculation of water flux. All membranes were dried at 70 °C overnight in an oven before permeation tests.

The desalination performance was evaluated in terms of water flux, which is defined as follows:
(2)$$Water\;flux\;(WF)=\frac{V}{At}$$
where *V* (L) is the volume of permeate, *A* (m^2^) is effective membrane area, *t* (h) the time of operation under transmembrane pressure (Δ*P*) = *P_f_* − *P_p_*= ~1atm. The ion rejection was calculated as follows:
(3)$$Ion\;rejection\;(\%)=\frac{C_{f}-C_{P}}{C_{f}}$$
where *C_f_* is the ion concentration in the feed side and *C_p_* is the ion concentration in the permeate side.

### 2.3. Membrane Characterization

Scanning Electron Microscope (SEM) micrographs were used to examine the growth of ZnO ALD on the membranes. FEI Quanta 600F field emission SEM (FEI, Hillsboro, OR, USA) attached with EDX Unit (Energy Dispersive X–ray Analyses) and EVEX nano-analysis System IV (EVEX Inc., Princeton, NJ, USA) were used to investigate the morphology and chemical elements of membrane surface and cross-section. The surface of the sample was scanned by accelerating a beam of fine focused electron under maximum potential difference of 20 kV. For SEM images, samples were first coated by gold layer (thickness of 200–300 Å) and then analyzed. For EDX analysis, samples were not coated and directly used for elemental analysis. In order to investigate the effect of fouling, both the feed and permeate side of the membranes were analyzed to observe the concentration of the retained ions before and after desalination tests. Elemental analysis by inductively coupled plasma (ICP) mass spectroscopy was used to measure the ion concentration in feed and permeate solution after desalination experiment. In order to measure the contact angle, DI water (2 µL) was dropped on the membrane surface at room temperature. Contact angles were measured using a goniometer (rame-hart model 250) between water–air interface and substrate surface. The left contact angle (θ_1_) and right contact angle (θ_2_) of the water drop (most of the time the value of θ_1_ and θ_2_ were close) were measured and the average of three contact angles measurements (each for left and right side) was taken to be the final contact angle. The same measurement was done at three different positions on the membrane.

## 3. Results and Discussion

### 3.1. ZnO ALD Membrane Characterization

Figure 3 shows the SEM micrographs of surface morphology for the α-alumina membrane before and after ZnO ALD. Before and after ZnO ALD, no significant change in membrane morphology was observed with SEM images. Figure 4 shows the EDX elemental mapping of the membrane before and after ZnO ALD. With successive ZnO ALD cycles (~2 Å/cycle), the ZnO film was found to be more continuous, uniform and dense. The EDX elemental mapping of the membrane surface showed the presence of Zn uniformly coated and distributed on the membrane surface. This indicates that the ZnO film was efficiently grown on the membrane surface without any apparent effect of diffusion limitations.

The content of Zn and Al were measured at different number of ALD cycles using EDX spectroscopy. The oxygen-free EDX composition analysis has a permeation depth of ~1 µm. The number of ZnO ALD cycles changed Zn/Al ratio. Figure 5a shows that the Zn/Al ratio on the membrane side is increased from 0 (pristine membrane) to 0.11 (after 128 ZnO ALD cycles). The elemental analysis shows that the Zn/Al ratio increased gradually after each successive ZnO ALD, which indicates the growth of ZnO film on the membrane surface during the ALD process. In Figure 5b, the mass gain of the ZnO ALD membrane was monitored with successive ZnO ALD cycles (4–128 cycles). With increasing ALD cycles, the mass of the ZnO ALD membrane increased, which is due to more ZnO deposition on the membrane. In the initial ALD cycles, ZnO is supposed to deposit as isolated particulates on the surface of the membrane. Further ALD cycles resulted in the outgrowth of ZnO particulates previously formed from nuclei, which might cover the membrane external surface and limit the probability for the precursor (diethylzinc) to enter the pores. Thus, in the higher number of ALD cycles (>20), the mass gain rate was relatively slower compared to that of the lower ALD cycle number (<20).

With increasing the number of ZnO ALD cycles, the membranes are expected to have increased affinity for water. In Figure 6, the hydrophilicity of the membranes before and after ZnO ALD was determined by measuring the water contact angle (CA) [32]. It was observed that the water CA continuously decreased from 101.2° to 93.8° as the number of ZnO ALD increased from 4 to 128, which indicated that ZnO ALD led to a progressively increased hydrophilicity of the membranes [33,34,35]. Hurwitz et al. investigated the impact of electrolyte pH, ionic strength and ionic concentration on surface acid-base functionality, wettability, and hydrophilicity. For buffered and unbuffered solutions water contact angle decreases from 61° to 42° and 72° to 46°, respectively, when the pH was increased from 2 to 12, which illustrates the increase in hydrophilicity with pH [35]. Ponsonnet et al. studied surface hydrophilicity, surface free energy, interfacial free energy and surface roughness for substratum surface. For titanium-nickel surface water contact angle decreased from 73° to 66° within initial 20 s which showed the increase in hydrophilicity [36].

### 3.2. ZnO ALD Membrane Performance

ZnO ALD membranes were used for the vacuum flow-through evaporation technique. Salty water passed through the ZnO ALD membrane in liquid form. Due to the vacuum in the permeate side, it reaches the evaporation state at room temperature, and the water on top of the membrane (permeate side) is evaporated. Over time, highly concentrated salty water was left on top of the permeate side of the membrane, as shown in Figure 2. In the case of pervaporation, the size of the membrane pore is relatively small (<10 nm) and water transports through the membrane as vapor [12]. However, in the vacuum flow-through evaporation, the size of the membrane pore is relatively larger (~200 nm), and there is a bulk movement of water across the membrane in liquid form.

Figure 7 shows detailed results for water flux and ion rejection rate for the pristine Al_2_O_3_ membrane and the ZnO ALD-treated membrane. The pristine membrane without ALD showed water flux of 117 L m^−2^ h^−1^, and it dropped to 92 L m^−2^ h^−1^ with successive desalination tests. This is because with successive desalination tests, membranes pores get covered with different salts from the salty feed water and it eventually leads to fouling and thus decreases membrane flux. In contrast, the ZnO ALD membrane showed gradual increase of water flux with increasing number of ALD cycles. With increasing the number of ALD cycles, the hydrophilicity of the membrane increased, which further causes the increase in water flux. The maximum water flux leveled off at 169 L m^−2^ h^−1^ after 128 ZnO ALD cycles. The water flux of the ZnO ALD membrane (169 L m^−2^ h^−1^) is found to be ~83% higher than that achieved by the pristine membrane (92 L m^−2^ h^−1^) after equal number of desalination experiment were performed. With increasing ZnO ALD cycles, the membrane surface became more hydrophilic, which is supported by the CA results [37]. However, ion rejection did not vary much by the ZnO ALD. For both the pristine membrane and ZnO ALD membranes, salt rejection values were >99.9%. This is because the desalination mechanism from vacuum-flow through evaporation is the same in both cases. Overall, the ZnO treated membrane exhibited very good salt rejection and better water flux than pristine membrane.

### 3.3. Effect of Ultrasonic Membrane Cleaning

Experiments were performed to investigate the effect of ultrasonic cleaning on membrane fouling, which is the major limitation in all membrane operations. To investigate the cleaning process, two pristine membranes were used for desalination experiments at the same conditions. After each cycle of desalination experiments, only one of the membranes was cleaned. After each step of desalination experiment, the membrane was detached from a bell-shaped glass tube. The tested membrane was calcined in stagnant air at 550 °C for 3.5 h to remove the remaining epoxy adhesive. The calcined membrane was immersed in an ultrasonication bath at room temperature for 30 min. Ultrasonic membrane cleaning uses high-frequency sound waves to agitate the aqueous solution which further acts on retained ion inside tortuous pores of the membrane surface. It was reported that sonication increased the cleaning efficiency by 5–10% [38]. After desalination experiments, cleaned and uncleaned membranes were analyzed with EDX in order to measure the retained ion concentrations on the surface and cross-section of the membranes. Figure 8a,b shows the results of EDX analysis of the membrane surface (feed side). The cleaned membrane showed much lower concentration of retained ion concentration (0.01–0.03%) compared to that of the uncleaned membrane (3–8%).

EDX elemental mapping (Figure 9a–d) showed that Ca^2+^ ion had lower ion concentration for cleaned membranes than for uncleaned membranes. Ca^2+^ ion concentration had a relatively small value of ~0.11 wt% for the membrane feed side. For the uncleaned membranes, the amount of salts on the surface got accumulated and the concentration of retained ion was higher (0.45 wt%) for the membrane feed side. It was found that cleaning process between desalination experiments helped in removing the ions retained on the membrane surface and inside the tortuous pore channels. Figure 9e–h also demonstrated that Na^+^ ion showed slightly higher concentration for the uncleaned membrane (0.15 wt% for membrane side) than for cleaned membrane (0.10 wt% for membrane side). The concentration was slightly higher for Ca^2+^ when compared to Na^+^. This can be explained by the higher hydrated ion radius of the Ca^2+^(0.41 nm) in comparison to Na^+^ (0.36 nm), which allowed more retention of Ca^2+^ ions in the tortuous pores and thus more concentration of Ca^2+^ in comparison to Na^+^ [39,40].

Because salty water flows across the membrane during desalination experiments, deposition of salt occurs in the tortuous pore channels along the membrane cross-section. To investigate this, we measured the spatial distribution of the metal ions along the cross-sections of the used membranes with EDX spectroscopy. Figure 10 shows the concentration profiles of Ca^2+^ and Na^+^ ions along the cross-sections of the cleaned and uncleaned membranes after seven desalination tests. Along the whole thickness of the membrane, it was observed that the cleaned membrane showed lower concentration for both Ca^2+^ and Na^+^ ions than the uncleaned membrane. This can be attributed to the cleaning procedure (both calcining and ultrasonication), which helped in removing the retained ions from the tortuous pores. Thus, membranes without cleaning showed higher concentrations of the retained ions. Compared to vacuum flow-through evaporation (170 L m^−2^ h^−1^), the water flux values obtained by conventional desalination methods were reported to be lower (~50 L m^−2^ h^−1^) [12]. Therefore, the vacuum assisted evaporation technique successfully achieved water flux almost three times higher than with the conventional desalination techniques. This implies that the vacuum flow-through evaporation technique can intensify desalination process.

## 4. Conclusions

ZnO was deposited on macroporous α-alumina membrane *via* atomic layer deposition (ALD) to improve water flux by increasing their hydrophilicity and reducing mass transfer resistance through membrane pore channels. It was observed from contact angles measurements that the membrane hydrophilicity was enhanced with increasing ZnO ALD cycles. Desalination with ZnO ALD membranes was conducted by a vacuum-assisted ‘flow-through’ evaporation method. Due to the high driving forces for water transport in the vacuum assisted method, there was bulk transport of water through the membrane, and then successive evaporation caused both extremely high water flux (~170 L m^−2^ h^−1^) and high purification of salty water (>99.5%). The vacuum-assisted ‘flow-through’ technique of the ZnO ALD membrane showed over three times higher water flux than those of pervaporation and membrane distillation. It was also shown that ultrasonic membrane cleaning had considerable impact on reducing the membrane fouling. By investigating the EDX spatial distribution of the metal ions along the cross-section and surface of the used membranes, the cleaned membranes showed lower concentration for both Ca^2+^ and Na^+^ ions than the uncleaned membranes. In combination with the ZnO ALD membrane, the vacuum-assisted ‘flow-through’ evaporation offers a superior technique for desalination and water purifications.

## Figures and Tables

**Figure 1 membranes-09-00156-f001:**
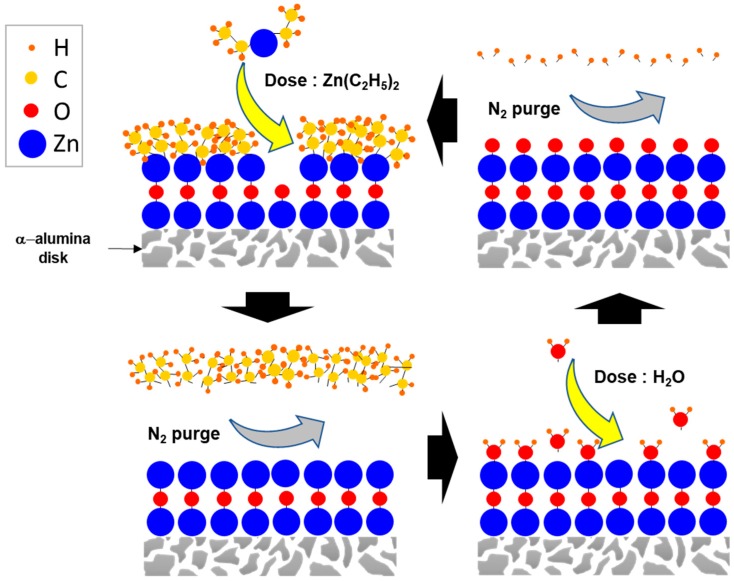
Schematic diagram of the atomic layer deposition (ALD) process in depositing ZnO on the surface of α-alumina membrane using diethylzinc and water precursors.

**Figure 2 membranes-09-00156-f002:**
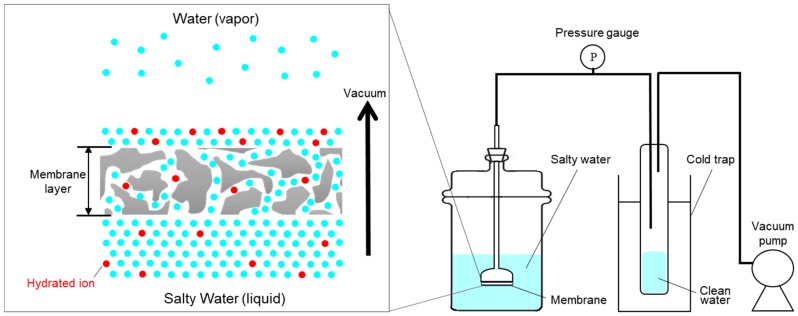
Schematic diagram of the experimental setup for the vacuum-assisted flow-through desalination experiment.

**Figure 3 membranes-09-00156-f003:**
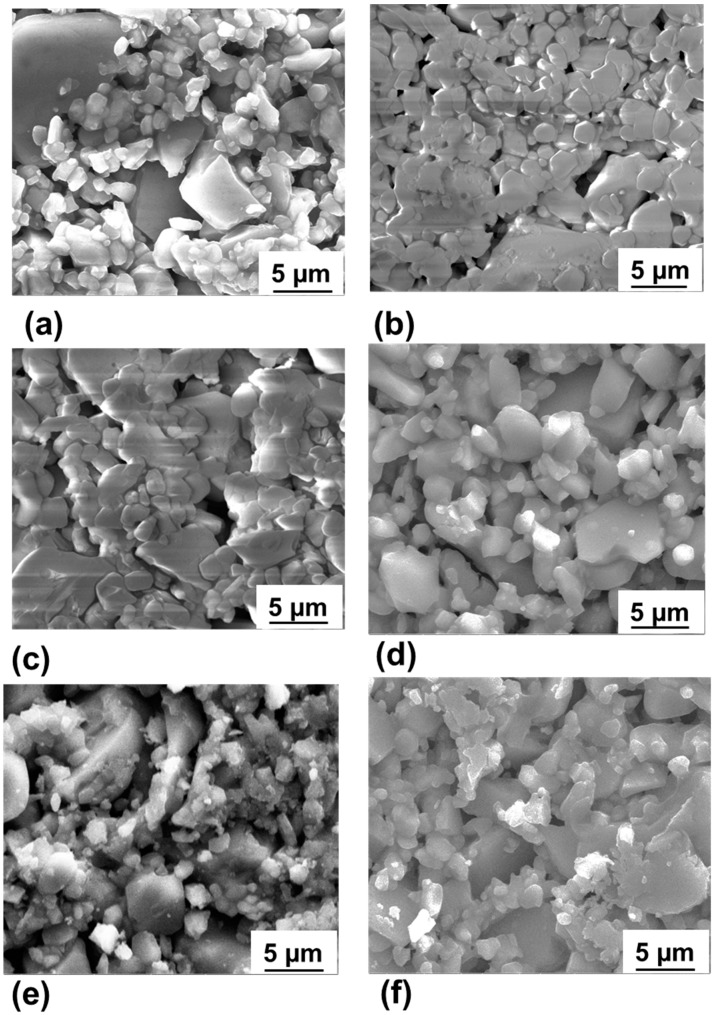
SEM images of (**a**) the pristine membrane and ZnO ALD membrane with (**b**) 8, (**c**) 16, (**d**) 32, (**e**) 64, and (**f**) 128 ALD cycles, respectively.

**Figure 4 membranes-09-00156-f004:**
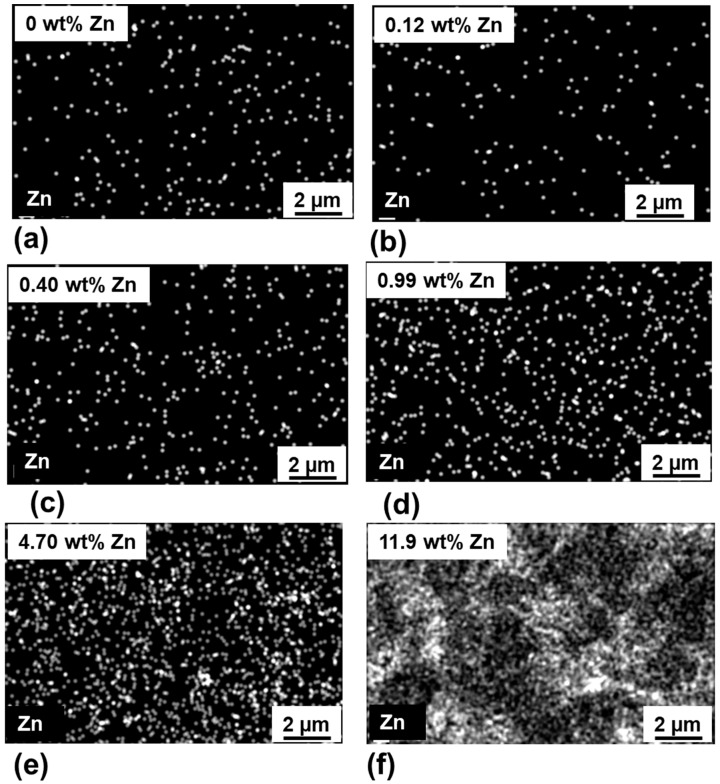
EDX elemental mapping of (**a**) the pristine membrane and ZnO ALD membrane with (**b**) 8, (**c**) 16, (**d**) 32, (**e**) 64, and (**f**) 128 ALD cycles, respectively.

**Figure 5 membranes-09-00156-f005:**
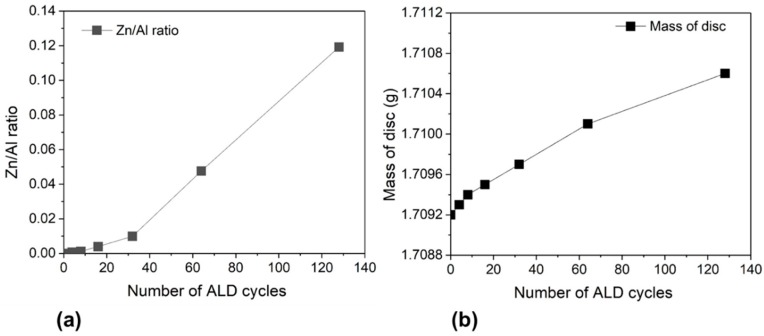
(**a**) Zn/Al content on the surface of the membrane and (**b**) variation of mass of membrane with the number of ZnO ALD cycles.

**Figure 6 membranes-09-00156-f006:**
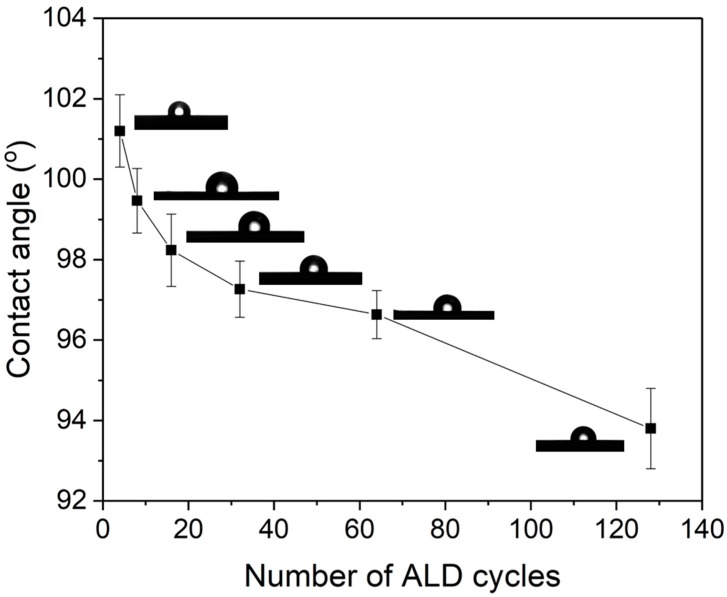
Water contact angle of α-alumina membrane with the number of ZnO ALD cycles.

**Figure 7 membranes-09-00156-f007:**
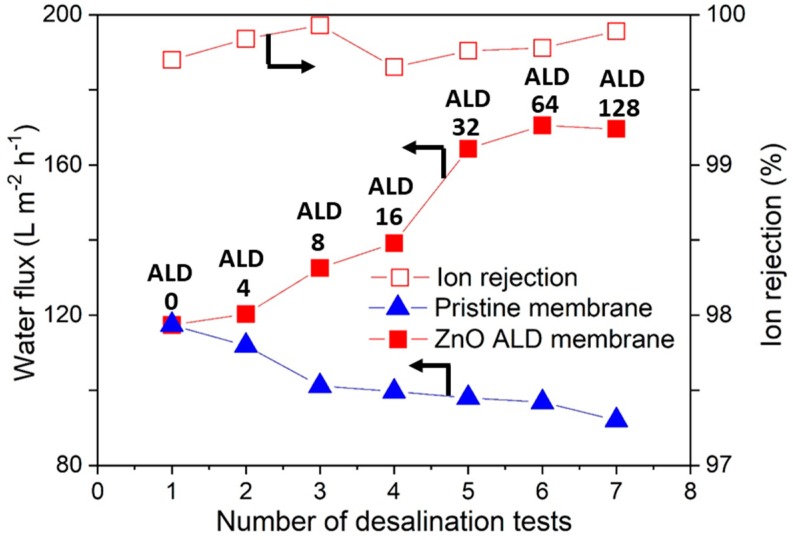
Water flux and ion rejection across the membrane with the number of ZnO ALD cycles.

**Figure 8 membranes-09-00156-f008:**
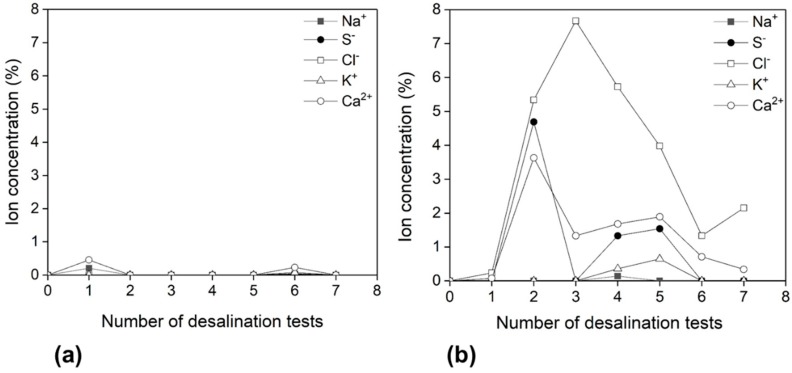
Ion concentration profiles on membrane surface (feed side) as a function of the number of desalination experiments for (**a**) membrane with cleaning and (**b**) membrane without cleaning in between desalination experiments.

**Figure 9 membranes-09-00156-f009:**
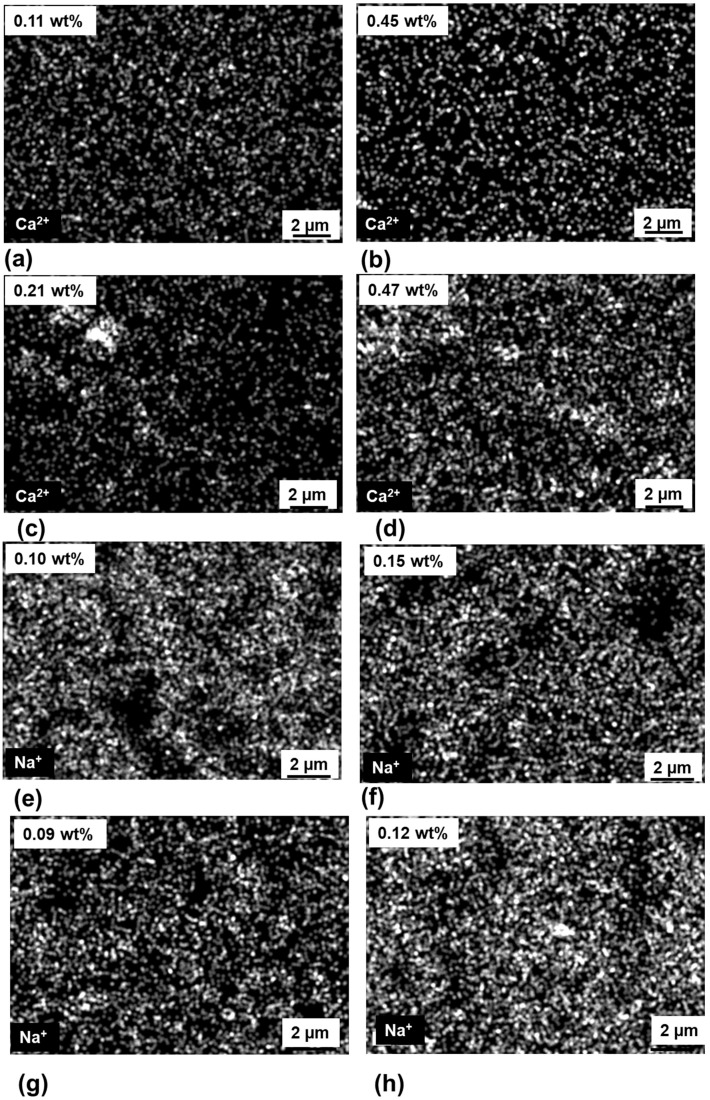
EDX elemental mapping of Ca^2+^ ion for membrane in the: (**a**) feed side with cleaning, (**b**) feed side without cleaning, (**c**) permeate side with cleaning, and (**d**) permeate side without cleaning and Na^+^ ion for membrane in the (**e**) feed side with cleaning, (**f**) feed side without cleaning, (**g**) permeate side with cleaning, and (**h**) permeate side without cleaning.

**Figure 10 membranes-09-00156-f010:**
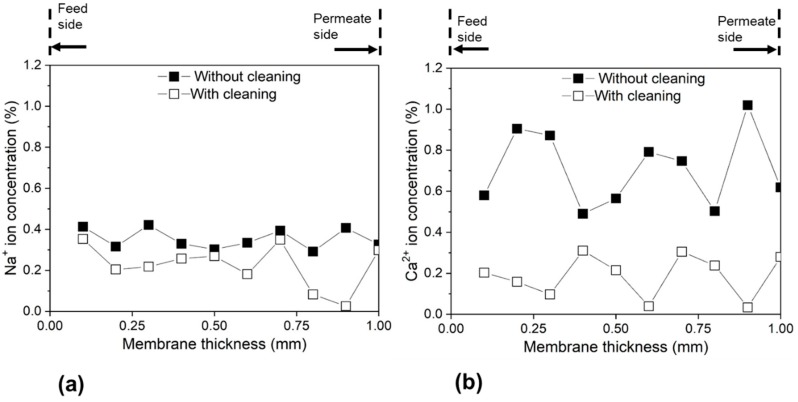
Ion concentration profiles along the membrane cross-section after seven desalination experiments with and without cleaning process: (**a**) Na^+^ and (**b**) Ca^2+^ ion.

**Table 1 membranes-09-00156-t001:** Literature review for salty water purification using membranes.

Methods	Membrane	Membrane Pore Size (nm)	Salt Concentration (ppm)	Ion Rejection (%)	Flux (L m^−2^ h^−1^)	T_feed_ (°C)	T_perm_ (°C)	Ref.
Pervaporation	SiO_2_	1.8	-	-	9.5	25	25	[7]
	SiO_2_	1.8–3.1	-	-	6.6	25	25	[14]
	CoO-SiO_2_	<10	-	-	7.7	25	25	[15]
	TiO_2_	3.1	-	-	6.0	25	25	[16]
	CMS-Al_2_O_3_	0.5–7.0	-	-	25	25	25	[17]
	Clinoptilolites zeolite	0.72	-	98.0	2.5	25	25	[18]
Membrane distillation	Polypropylene membrane	200	21,200	99.9	11	60	20	[11]
	Electrospun membrane	1250	10,000	99.9	8.5	65	25	[19]
	SiO_2_ composite membrane	703	35,000	-	23	70	25	[20]
	PVDF Mixed matrix membrane	970	35,000	99.9	6.2	80	25	[10]

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
