# Peer review of "ZnO Microfiltration Membranes for Desalination by a Vacuum Flow-Through Evaporation Method"

_membranes, 2019, doi:10.3390/membranes9120156_

Round 1

Reviewer 1 Report

639149-Membranes review: Title: ZnO Microfiltration Membranes for Desalination by a Vacuum Flow-Through Evaporation Method

I have finished reviewing the manuscript submitted for publication in Membranes. The overall suggestion I have is that the paper is acceptable for publication after minor revision. The manuscript contains new and valuable results and is worthy to be published. The manuscript “ZnO Microfiltration Membranes for Desalination by a Vacuum Flow-Through Evaporation Method” shows that ZnO could increase the membrane hydrophilicity and reduce the mass transfer resistance through membrane pore channels by ALD cycles.

The paper is written in quite good English.

My specific comments and questions are as follows:

In Equation 2: The TMP, transmembrane pressure should be in it. What was the real value of it? What were the measuring errors of the contact angle measurements? In Figure 6: The measuring errors can be shown! How many parallel experiments were carried out? The VRR: volume reduction ratios must be given at the end of the experiments. What was the concentration ratio? From figure 7: more details should be explained in the text, because it has/shown very valuable results.

Reviewer 2 Report

The manuscript reports on  ZnO Microfiltration Membranes for Desalination by  a Vacuum Flow-Through Evaporation Method. The work is clearly presented and the performance of the device appears to be satisfactory. Minor revision is recommended, e.g.

      Lines 35-… desalination option and improve the economics….. R: …improves….

145 - where Cf is the ion concentration in the feed side and Cp is the ion concentration in the permeate. R:  Italics

280- with EDX spectroscopy. Figure 10 shows the concentration profiles of Ca2+and Na+ …. R: Space between Ca2+ and  Na+….
